# Molecular Epidemiology of Carbapenem-Resistant *P. aeruginosa* and *Enterobacterales* Isolates from Clinical Infections and Their Susceptibility to Ceftazidime–Avibactam

**DOI:** 10.3390/microorganisms13092015

**Published:** 2025-08-29

**Authors:** Jose-Rita Gerges, Sara Barada, Hadi Hussein, Ahmad Sleiman, Ziad Jabbour, Fatima El Rida, Abdallah Kurdi, Ghassan Matar, George Araj, Antoine Abou Fayad, Zeina Adnan Kanafani

**Affiliations:** 1Department of Experimental Pathology, Immunology, and Microbiology, Faculty of Medicine, American University of Beirut, Beirut 1107 2020, Lebanon; jg39@aub.edu.lb (J.-R.G.); sb133@aub.edu.lb (S.B.); hh161@aub.edu.lb (H.H.); ahmad.sleiman@uclouvain.be (A.S.); zj20@aub.edu.lb (Z.J.); fae34@aub.edu.lb (F.E.R.); gmatar@aub.edu.lb (G.M.); aa328@aub.edu.lb (A.A.F.); 2Center for Infectious Diseases Research, American University of Beirut, Beirut 1107 2020, Lebanon; 3World Health Organization (WHO) Collaborating Center for Reference and Research on Bacterial Pathogens, Beirut 1107 2020, Lebanon; 4Department of Biochemistry and Molecular Genetics, Faculty of Medicine, American University of Beirut, Beirut 1107 2020, Lebanon; ak161@aub.edu.lb; 5Department of Pathology and Laboratory Medicine, American University of Beirut Medical Center, Beirut 1107 2020, Lebanon; garaj@aub.edu.lb; 6Department of Internal Medicine, American University of Beirut Medical Center, Beirut 1107 2020, Lebanon

**Keywords:** ceftazidime/avibactam, *Pseudomonas aeruginosa*, *Enterobacterales*, antimicrobial resistance, metallo-β-lactamase, resistance mechanisms, efflux pumps

## Abstract

The overuse of carbapenems has driven the rise of carbapenem-resistant *Pseudomonas aeruginosa* (CRPA) and *Enterobacterales* (CRE), against which ceftazidime–avibactam (CAZ-AVI) offers an alternative treatment. This study phenotypically determined resistance profiles of *P. aeruginosa* (PA), *Escherichia coli* (EC), and *Klebsiella pneumoniae* (KP) clinical isolates to CAZ-AVI and investigated molecular resistance mechanisms genotypically. A total of 394 PA, 90 EC, and 84 KP isolates were collected from the American University of Beirut Medical Center. Antimicrobial susceptibility testing (AST) and whole-genome sequencing (WGS) were performed on 30 isolates per species. Results showed that 100% of KP, 63% of EC, and 100% of PA isolates were carbapenem-resistant. Among these, 73% of KP, 79% of EC, and 60% of PA were CAZ-AVI-resistant. WGS revealed diverse sequence types, plasmids, and antimicrobial resistance genes. Additionally, 100% of KP, 93% of EC, and 89% of PA isolates produced metallo-β-lactamases (MBLs). Mutations in *ampD*, *ampR*, and *mexR* were identified in CAZ-AVI-resistant, non-MBL-producing PA, whereas mutations in *ompC*, *marR*, and *ampC* were detected in CAZ-AVI-resistant, non-MBL-producing EC. While CAZ-AVI remains effective against most CRE and CRPA lacking MBLs, resistance to CAZ-AVI is multifactorial, commonly involving overexpression of efflux pumps and AmpC β-lactamases, loss of porin channels, and the presence of oxacillinases.

## 1. Introduction

According to the World Health Organization (WHO) and Centers for Disease Control and Prevention (CDC), multidrug-resistant Gram-negative bacteria are a serious threat to public health, affecting the ability to treat infections and increasing morbidity, mortality, and healthcare costs [1,2]. A recent systematic analysis of the global burden of antimicrobial resistance estimated that 4.95 million deaths were associated with bacterial AMR in 2019 worldwide, making it one of the leading global health threats [1]. Moreover, the misuse and overuse of antibiotics has contributed to the acceleration of antimicrobial resistance worldwide [3].

Due to the limited number of new antimicrobial agents, the WHO has published, since 2017, a priority list of novel antibiotics for the development of effective treatments against those pathogens considered critical, such as carbapenem-resistant *A. baumannii*, carbapenem-resistant *Pseudomonas aeruginosa*, and carbapenem-resistant and ESBL-producing *Enterobacteriaceae* [4]. In the Middle East and North Africa (MENA) region, surveillance reports consistently document high rates of carbapenem resistance, often exceeding 50% for *K. pneumoniae* and *P. aeruginosa*, with blaNDM and blaVIM being the most prevalent carbapenemase genes [5]. In Lebanon specifically, outbreaks of NDM-producing *Enterobacterales* and VIM-producing *P. aeruginosa* have been reported in multiple hospitals, reflecting a sustained circulation of these high-risk clones [6]. Infections caused by such isolates are associated with worse clinical outcomes than infection caused by susceptible strains. In fact, meta-analyses have reported high mortality rates for CRE bacteremia and CRPA bloodstream infections [7,8]. These infections are also linked to prolonged hospital stays and markedly higher healthcare costs due to the need for combination or last-line therapies, intensive care management, and isolation precautions [9]. Such outcomes place a significant burden on healthcare systems, particularly in limited resources settings.

Carbapenems belong to the beta-lactam antibiotics family and are used to provide broad-spectrum coverage against various types of bacteria [10]. Over the past 30 years, the emergence of ESBL-producing *Enterobacteriaceae* has been observed [11], leading to a significant increase in the use of carbapenems to treat infections caused by these organisms [12].

Unfortunately, this increased utilization of carbapenems has contributed to the development of resistance to these antibiotics in *Pseudomonas aeruginosa* (CRPA) and *Enterobacterales* (CRE) through various mechanisms such as OXA-48-like family (class D Oxacillinases) with carbapenem-hydrolizing activity, *Klebsiella pneumoniae* carbapenemases (KPCs), New Delhi Metallo-β-lactamase (NDM), Imipenemase (IMP), and Verona integron-encoded metallo-β-lactamase (VIM). Local multicenter surveillance studies in the region report the steady increase of carbapenem-resistant *K. pneumonia*, *E. coli*, and *P. aeruginosa* over the past decade [5]. Additionally, resistance can occur through the loss of porins and up-regulation of efflux pumps [13,14].

Several studies have investigated the risk factors associated with the acquisition of infections caused by multidrug-resistant *P. aeruginosa* and *Enterobacterales*. A study showed that infection with CRE was associated with long-term hospitalization, major surgery, and broad-spectrum antibiotic therapy [15].

CAZ-AVI is a novel β-lactam/β-lactamase inhibitor combination with activity against *Pseudomonas aeruginosa* and carbapenem-resistant *Enterobacterales*, except the ones carrying metallo-β-lactamases [16]. Avibactam inhibits Ambler class A, class C, and some class D β-lactamases, restoring ceftazidime activity against organisms producing these enzymes. [17]. Surveillance studies have shown high CAZ-AVI susceptibility rates among KPC- and OXA-48-producing isolates [18]. However, its effectiveness is often threatened by both the resistance conferred by MBL-producing strains and porin alterations combined with carbapenemase gene mutation [19,20]. Despite this, CAZ-AVI remains an important option for treating infections caused by KPC- and OXA-48-producing isolates and can help reduce carbapenem use, potentially slowing resistance development [21]. Therefore, it is crucial to determine phenotypically the resistance profiles of *Pseudomonas aeruginosa*, *Escherichia coli*, and *Klebsiella pneumoniae* clinical isolates against CAZ-AVI and investigate genotypically the molecular mechanisms of resistance.

## 2. Materials and Methods

### 2.1. Bacterial Isolates

A total of 394 *P. aeruginosa*, 90 *E. coli*, and 84 *K. pneumoniae* isolates, collected between 2017 and 2024, were delivered from the Pathology and Laboratory Medicine department at the American University of Beirut, Medical Center to the bacteriology and molecular microbiology laboratory at the American University of Beirut.

### 2.2. Antimicrobial Susceptibility Testing

Antimicrobial susceptibly testing (AST) was performed using both the disc diffusion and broth microdilution methods against different antimicrobials from different antimicrobial classes, with a focus on ceftazidime–avibactam (CAZ-AVI), meropenem–vaborbactam (MEV), and imipenem–relebactam (IMR). Results were interpreted according to the CLSI M100 guideline Ed 35 [22].

The choice of method was based on resource availability and the suitability of the method for specific antibiotics. Disk diffusion was used as the standard approach for the majority of antibiotics due to its cost-effectiveness and accessibility in our routine laboratory setting. The broth microdilution method was employed for antibiotics, such as colistin, where disk diffusion is not reliable or not recommended due to poor diffusion characteristics. No formal comparison or correlation analysis was performed between the two methods, as they were not applied in parallel for all agents but rather in a complementary manner.

### 2.3. Disk Diffusion

For each isolate, a bacterial suspension equivalent to 0.5 MacFarland was prepared. Then, it was subcultured on a round Mueller–Hinton agar plate, in all the directions, to ensure that the bacterial suspension covered all the plate, using a sterile swab. The plate was left for around 10 min closed on the bench, followed by the addition of the tested antimicrobials. The plate was then incubated at 37 °C for 18–24 h after which the zone of inhibition diameters was measured.

### 2.4. Broth Microdilution

Serial dilution was performed with concentrations ranging from 512 µg/mL to 0.5 µg/mL and the plate was incubated at 37 °C for 18–24 h. All the experiments were run in duplicates. The results were interpreted according to the CLSI M100 guideline. Control strain *P. aeruginosa* ATCC^®^ 27853, *E. coli* ATCC^®^ 25922, and *K. pneumoniae* ATCC^®^ 13833 were used in parallel to monitor the MIC results.

### 2.5. Resistance Classification

Multidrug-resistant (MDR), extensively drug-resistant (XDR), and pandrug-resistant (PDR) classifications were defined according to the international consensus criteria proposed by Magiorakos et al. [23], using susceptibility profiles interpreted based on CLSI breakpoints. MDR was defined as non-susceptibility to at least one agent in three or more antimicrobial categories; XDR as non-susceptibility to at least one agent in all but two or fewer categories; and PDR as non-susceptibility to all agents in all categories. Antimicrobial categories were selected based on the list provided in the original publication and adapted to the antibiotics tested in our study, ensuring coverage of the major classes outlined in the consensus definitions.

### 2.6. Whole-Genome Sequencing

A total of 30 *P. aeruginosa*, 30 *E. coli*, and 30 *K. pneumoniae* were sequenced. The selection was based on phenotypic diversity, including a range of antimicrobial resistance profiles (from susceptible to MDR, XDR) and unique resistance patterns that merited further genomic investigation. This selection aimed to reflect the broader collection while also considering feasibility and cost limitations.

To prepare whole-genome sequencing libraries, fresh cultures were grown on LB and MacConkey agar and genomic DNA was extracted using the Quick-DNA™ Fungal/Bacterial Miniprep kit (Zymo Research, Irvine, CA, USA); then, DNA was purified using the DNA Cleanup and Concentrator (Genomic DNA Clean & Concentrator™) kit (Zymo Research) according to the manufacturer’s protocols. Sequencing libraries were prepared using the Illumina DNA prep kit (Illumina, San Diego, CA, USA) and sequenced on an Illumina MiSeq sequencer, 2 × 250 cycles.

### 2.7. Bioinformatics Analysis

Reads quality control and trimming was performed using FastQC and Trimmomatic (v.1.2.14) after which assembly of the genome was performed using Unicycler on Galaxy “https://usegalaxy.org/ (accessed on 20 May 2025)”.

Antimicrobial resistance genes were acquired through the Comprehensive Antibiotic Resistance Database (CARD) “https://card.mcmaster.ca/ (accessed on 20 May 2025)”.

Plasmids harbored in each isolate were determined using PlasmidFinder from the Center of Genomic Epidemiology (CGE) “https://cge.food.dtu.dk/services/PlasmidFinder/ (accessed on 20 May 2025)”.

Sequence types were assigned using multilocus sequence typing (MLST) schemes specific to each species via the MLST tool from the CGE. For *K. pneumoniae*, the scheme included the housekeeping genes gapA, infB, mdh, pgi, phoE, rpoB, and tonB. For *E. coli*, the genes analyzed were adk, fumC, gyrB, icd, mdh, purA, and recA, as per the Achtman MLST scheme. For *P. aeruginosa*, we used the standard scheme targeting acsA, aroE, guaA, mutL, nuoD, ppsA, and trpE. Assignments were cross-validated using the PubMLST database “https://cge.food.dtu.dk/services/MLST/ (accessed on 20 May 2025)”.

R package called “Complex Heatmap” (2.18.0) was used to illustrate 3 different heatmaps based on their resistance gene profiles and sequence types.

### 2.8. Variant Calling

Sequencing quality reports were generated for the raw FASTQ files using FastQC (v0.12.1) to check for low-quality bases and presence of adapters. After that, the VariantDetective (PMID: 38366603) pipeline was used to call variants. Briefly, the pipeline aligns the raw reads to the reference genome pseudomonas_aeruginosa_pao1 and escherichia_coli_str_k_12_substr_mg1655_gca_000005845 downloaded from Ensembl using BWA (PMID: 19451168). Aligned reads are stored in sequence alignment map (SAM) files for each isolate. Then, consensus single nucleotide polymorphisms (SNPs) and short insertions and deletions (INDELs) are called using Freebayes “https://arxiv.org/abs/1207.3907 (accessed on 21 May 2025)”, GATK HaplotypeCaller (PMID: 20644199), and Clair3 (PMID: 38177392) tools using the SAM files generated from the previous step as input. The final output is a VCF file that contains the consensus variant calls. Ensembl-VEP (PMID: 27268795) was run to annotate the called variants and a tabulated form of the variants was generated using bcftools (PMID: 33590861) + split-vep plugin.

## 3. Results

### 3.1. Antimicrobial Susceptibility Testing Results

Our results showed that 37 out of 84 *K. pneumoniae* isolates (44%) were MDR, 35 (42%) were XDR, and 11 (13%) were PDR.

Moreover, 82 out of 90 *E. coli* isolates (91%) were MDR and 5 (6%) were XDR.

Furthermore, 115 out of 394 *P. aeruginosa* isolates (29%) were MDR, 43 (11%) were XDR, and 1 (0.3%) was PDR.

AST results showed varying resistance rates for the different bacterial species against different antimicrobials. The highest resistance rates observed for *K. pneumoniae* and *E. coli* were against cefuroxime (99% and 98%, respectively) and for *P. aeruginosa* against levofloxacin (66%). On the other hand, the lowest resistance rates observed for *K. pneumoniae* and *E. coli* were against amikacin (52% and 7%, respectively) and for *P. aeruginosa* against ceftazidime–avibactam (18%) (Figure 1).

Furthermore, 90% of *K. pneumoniae,* 70% of *E. coli*, and 31% of *P. aeruginosa* isolates were found to be resistant to meropenem, while resistance to CAZ-AVI was observed in 65%, 48%, and 18% of these isolates, respectively.

In addition, lower resistance rates against CAZ-AVI (65%, 48%, and 18%) compared to ceftazidime (93%, 84%, and 24%) were detected in *K. pneumoniae*, *E. coli*, and *P. aeruginosa* isolates, respectively (Figure 2).

### 3.2. Whole-Genome Sequencing (WGS) Analysis

To investigate the molecular mechanism of resistance, WGS was conducted on 30 isolates from each bacterial species.

#### 3.2.1. Sequence Types (STs)

Among the sequenced isolates, 13, 17, and 8 different STs were observed for each *K. pneumoniae*, *E. coli*, and *P. aeruginosa* isolate, with ST383 (47%), ST648 (30%), and ST111 (50%) being the most frequent, respectively. Interestingly, 7% of *K. pneumoniae* and *E. coli* and 3% of *P. aeruginosa* isolates were of unknown ST (Figure 3).

#### 3.2.2. Plasmids

*K. pneumoniae* isolates harbored 19 different plasmids, with IncHI1B (pNDM-MAR) (63%) and IncFIB (pNDM-Mar) (47%) being the most frequent. Regarding *E. coli* isolates, 31 different plasmids were detected, among which IncFIA (60%), IncFIB (AP001918) (53%), and IncFII (43%) were the most commonly observed.

#### 3.2.3. AMR Genes

WGS analysis also shows that *K. pneumoniae, E. coli*, and *P. aeruginosa* isolates encode numerous antimicrobial resistance determinants (97, 120, and 119, respectively).

The heatmap below (Figure 4) visualizes the distribution of various beta-lactamase genes, efflux pump components, and porins across *K. pneumoniae*, *E. coli*, and *P. aeruginosa* isolates, revealing their distinct genetic resistance landscapes and co-occurrence patterns with specific sequence types.

### 3.3. Antimicrobial Susceptibility and Molecular Characterization of CAZ-AVI-Resistant P. aeruginosa and Enterobaterales

Moreover, the rates of resistance to meropenem among sequenced isolates were as follows: 30 out of 30 (100%) for *K. pneumoniae,* 19 out of 30 (63%) for *E. coli*, and 30 out of 30 (100%) for *P. aeruginosa* isolates. Among those, 22 out of 30 (73%) *K. pneumoniae,* 15 out of 19 (79%) *E. coli*, and 18 out of 30 (60%) *P. aeruginosa* isolates were resistant to ceftazidime–avibactam (Figure 5). Furthermore, the percentages of isolates harboring metallo-β-lactamase genes among sequenced carbapenem and CAZ-AVI-resistant isolates were as follows: 22 out of 22 for *K. pneumoniae* (100%)*,* 14 out of 15 for *E. coli* (93%), and 16 out of 18 for *P. aeruginosa* (89%). Additionally, 15 out of 22 *K. pneumoniae* (68%)*,* 9 out of 15 *E. coli* (60%), and 18 out of 18 *P. aeruginosa* isolates (100%) were oxacillinase producers, of which 55% (12 out of 22) of *K. pneumoniae* isolates encoded carbapenemases (*bla_OXA-48_*, *bla_OXA-232_*) (Figure 6). Notably, some *K. pneumoniae* isolates harbored both MBL and OXA-type carbapenemases, indicating co-occurrence of multiple carbapenemase genes within the same strain.

### 3.4. WGS Analysis of Non-MBL CAZ-AVI-Resistance Isolates and Associated Resistance Genes

As no metallo-β-lactamase (MBL) genes were detected in the remaining two CAZ-AVI-resistant *P. aeruginosa* and *E. coli* isolates, resistance is likely driven by alternative mechanisms. To comprehensively elucidate these molecular bases of CAZ-AVI resistance in non-MBL-producing clinical isolates, we conducted detailed genetic analyses. This investigation focused on identifying mutations potentially leading to altered gene expression—either overexpression or repression—or amino acid substitutions that affect protein function. Specifically, for *P. aeruginosa*, we analyzed beta-lactamase-encoding gene regulators (*bla_AmpD_*, *bla_AmpR_*, *bla_AmpG_*), genes encoding the MexAB-OprM multidrug efflux pump regulators (*mexR*, *nalC*, *nalD*), and the *dacB* gene, which encodes penicillin-binding protein 4 (PBP4). However, for *E. coli,* we focused on beta-lactamase-encoding genes (*bla_AmpC_*), genes encoding the AcrAB-TolC efflux pump regulators (*mexR*), and genes encoding a major outer membrane porin *ompC*.

We found single nucleotide variants (SNVs) in *ampD* (D183Y, G148A, R11L) and *ampR* (D135N) for CAZ-AVI resistant PSA_202307_695. As for PSA_202309_721 isolate, SNVs were detected in *mexR* (V126E) and in *ampD* (A96T, E68D); in *ampR*, SNVs (M288R, S179T, E114A) and substitutions (RG282RE), all resulting in missense mutations contributing to phenotypic resistance to CAZ-AVI. For the *E. coli* isolate, we identified various amino acid alterations: in *ampC*, SNVs (D367A, D304G, L257R, A157T) and substitutions (RP312HP, NE260ND, LK254MN, VQ250VR, PPA208PPP); in *marR*, SNVs (G103S, Y137H), all resulting in missense mutations; in *ompC*, substitutions (SLA295SVA, GAI216AAV, GNPSG175GSVSG) resulting in missense mutations, and highly disruptive INDELs (RG308TIAGRN, FTSGVT182MT) leading to frameshift mutations (Appendix A). This finding suggests a possible link between the observed genotype and CAZ-AVI resistance.

## 4. Discussion

In this study, we determined the phenotypic resistance profiles of *K. pneumoniae*, *E. coli*, and *P. aeruginosa* clinical isolates, particularly against CAZ-AVI, and investigated the underlying molecular mechanism of resistance through WGS. We analyzed the susceptibility to CAZ-AVI in 394 *P. aeruginosa*, 90 *E. coli*, and 84 *K. pneumoniae* isolates collected between 2017 and 2024 in Lebanon. Finally, we performed WGS on 30 isolates from each species in order to uncover the antimicrobial resistance determinants associated with CAZ-AVI resistance.

Our findings show varied resistance rates against commonly used antimicrobials. In particular, 65% of *K. pneumoniae*, 48% of *E. coli*, and 18% of *P. aeruginosa* were CAZ-AVI-resistant. Moreover, avibactam is a β-lactamase inhibitor effective against class A, C, and D β-lactamases. When combined with ceftazidime, it prevents its hydrolysis by binding to the active site of serine-β-lactamases, thus restoring ceftazidime activity against resistant Gram-negative bacteria [19]. This is reflected in our findings where the resistance rate to CAZ-AVI was lower than to ceftazidime alone in all three species (65 vs. 93%, 48 vs. 84%, and 18 vs. 24%, respectively), highlighting the effectiveness of avibactam in restoring ceftazidime antibacterial activity [24].

Notably, a significant proportion of sequenced isolates exhibited carbapenem resistance: 100% of *K. pneumoniae*, 63% of *E. coli*, and 100% of *P. aeruginosa*. Among these, CAZ-AVI resistance rates were 73%, 79%, and 60%, respectively, underscoring concerns about CAZ-AVI’s effectiveness against multidrug-resistant Gram-negative bacteria and emphasizing the urgent need for novel antimicrobials [25,26]. Similar studies from regions with high MBL prevalence such as South Asia and North Africa have reported similarly high CAZ-AVI resistance rates when compared to areas with KPCs endemic such as the United States and Greece [27]. In fact, this geographical variability highlights the critical influence of local carbapenemase epidemiology on CAZ-AVI effectiveness and underscores the need for region-specific treatment guidelines and stewardship policies [27].

Moreover, using WGS, we uncovered the molecular mechanisms of resistance of these isolates by revealing their resistomes. In *Enterobacterales*, the predominant mechanism associated with CAZ-AVI resistance was the presence of metallo-β-lactamase (MBL)-encoding genes, particularly *bla_NDM_* variants. All 22 CAZ-AVI-resistant *K. pneumoniae* (100%) and 14 out of 15 CAZ-AVI-resistant *E. coli* isolates (93%) harbored *bla_NDM_* genes, which aligns with previously reported data indicating that MBL production is the leading cause of CAZ-AVI resistance [6]. In fact, avibactam does not inhibit Ambler class B β-lactamases, which are metal-dependent enzymes. Studies have shown that the active sites of these MBLs accommodate and hydrolyze ceftazidime regardless of the presence of avibactam [24], explaining the intrinsic resistance observed. Moreover, co-production of ESBLs or AmpCs with mutations leading to porin loss and/or efflux pump overexpression, further compounding treatment challenges. Among CAZ-AVI-resistant *P. aeruginosa* isolates, 89% were found to be MBL-producers, particularly *bla_VIM_* and *bla_IMP_* variants, which supports previous findings that the presence of MBLs in *P. aeruginosa* is a major contributor to CAZ-AVI resistance [28].

Our findings align with previous reports from Lebanon and the Middle East, where *bla_NDM_* was the most predominant MBL in carbapenem-resistant *Enterobacterales* and *bla_VIM_* in *Pseudomonas aeruginosa* [29]. For example, Sobh et al. documented the spread of NDM-5 as the most common carbapenemase in *Enterobacterales* while Al-alaq et al. reported VIM-1-carrying *P. aeruginosa* isolated from Lebanon. Moreover, similar patterns were reported in Bahrain, Saudi Arabia, and Egypt, where NDM-producing *K. pneumonia* and VIM-producing *P. aeruginosa* dominate the carbapenem-resistant Gram-negative landscape [30,31,32]. These trends are mirrored in our study where CAZ-AVI-resistance is due to MBL expression. This consistent pattern across studies highlights the sustained circulation of these high-risk carbapenemase genes in the whole region.

As no MBL genes were detected in the remaining CAZ-AVI-resistant *P. aeruginosa* and *E. coli* isolates, a detailed genomic analysis was conducted, revealing that CAZ-AVI resistance is multifactorial and attributed to other mechanisms. In *P. aeruginosa* isolates, resistance is likely driven by mutations in *ampD* and *ampR*, leading to AmpC hyperproduction [33], and in *mexR*, causing overexpression of the MexAB-OprM efflux pump, which slightly raises the MIC of CAZ-AVI [34,35,36]. For the *E. coli* isolate, resistance is primarily due to *ompC* frameshift INDELs severely reducing outer membrane permeability, a critical factor given CAZ-AVI’s reliance on passive diffusion for cellular entry [37]. Additionally, *marR* mutations enhance AcrAB-TolC efflux, and numerous *ampC* substitutions potentially alter beta-lactamase activity. These combined chromosomal alterations provide a robust mechanism for evading CAZ-AVI activity [38]. Such chromosomal mechanisms are often under-recognized in routine diagnostic workflows, which only focus on detecting acquired carbapenemase genes. Whole-genome sequencing therefore provides an invaluable tool for uncovering such hidden resistance mechanisms explaining therapeutic failures in non-carbapenemase-producing CAZ-AVI-resistant isolates [36].

Oxacillinase genes were also detected in 68% of CR-*K. pneumoniae*, 60% of CR-*E. coli*, and 100% of CRPA isolates. Although oxacillinases generally have a limited impact on CAZ-AVI activity, a subset of these genes encodes carbapenemases such as *bla_OXA-48_* and *bla_OXA-232_*, which are generally susceptible to CAZ-AVI when present alone, but may contribute to reduced susceptibility when combined with additional resistance mechanisms such as porin loss, efflux pump overexpression, or co-production of ESBLs or AmpC β-lactamases [39].

Our WGS results demonstrate a significant clonal diversity in addition to high numbers of AMR genes and plasmids. The most prominent STs identified were ST383 in *K. pneumoniae*, ST648 in *E. coli*, and ST111 in *P. aeruginosa*, known for their role in the global spread of multidrug resistance [40,41,42]. Moreover, the variety and high prevalence of AMR genes observed across our isolates highlights an alarming layered complexity and redundancy of resistance mechanisms, prompting urgent improvement of surveillance and control strategies on the national level. Furthermore, the presence of multiple plasmids, such as IncHI1B(pNDM-MAR) and IncFIB(pNDM-Mar) in *K. pneumoniae*, widely associated with *bla_NDM_* and *bla_OXA-48-like_* carbapenemase genes, suggests their role in the dissemination of carbapenem resistance. Moreover, IncFIA and IncFIB (AP001918) present in *E. coli* were frequently found in isolates carrying *bla_NDM_*, suggesting a potential for horizontal gene transfer leading to the spread of carbapenem-resistance genes [43,44].

Overall, the concordance between our genotypic and phenotypic data in *Enterobacterales* highlights NDM-type MBL production as the primary molecular etiology of CAZ-AVI resistance [45]. In contrast, we suggest that *P. aeruginosa* presents a more complex resistance profile in the absence of MBLs, where other resistance mechanisms, such as efflux pump overexpression and AmpC deregulation, potentially play a significant role in conferring CAZ-AVI resistance [25,46]. Nonetheless, the high frequency of CAZ-AVI resistance among isolates carrying MBLs and carbapenemases reaffirms the need for routine genotypic screening alongside phenotypic testing, especially in clinical settings where CAZ-AVI is considered a frontline treatment. From a therapeutic standpoint, CAZ-AVI remains an important option against non-MBL carbapenemase producers, particularly KPC- and OXA-48-like strains. For confirmed MBL producers, combination therapy with CAZ-AVI plus aztreonam offers a rational approach, targeting MBL producers as well as co-producers of serine β-lactamases [47]. Other agents such as cefiderocol and taniborbactam show in vitro activity against MBLs and are entering clinical use, overcoming MBL-producing isolates’ resistance to CAZ-AVI [48,49].

In conclusion, our findings reveal alarmingly high levels of resistance to CAZ-AVI among carbapenem-resistant *K. pneumoniae*, *E. coli*, and *P. aeruginosa* isolates. Whole-genome sequencing revealed a high diversity of sequence types and a wide variety of acquired resistance genes contributing to both carbapenem and ceftazidime–avibactam resistance. These results emphasize the critical importance of molecular diagnostics, antimicrobial stewardship, and collaborative surveillance to guide effective therapeutic strategies and mitigate the spread of CAZ-AVI resistance in high-burden settings. In the short term, combining other antimicrobial agents such as aztreonam with CAZ-AVI, or using alternative treatments such as cefiderocol, may offer a viable option against resistant isolates until novel antimicrobials are developed.

## Figures and Tables

**Figure 1 microorganisms-13-02015-f001:**
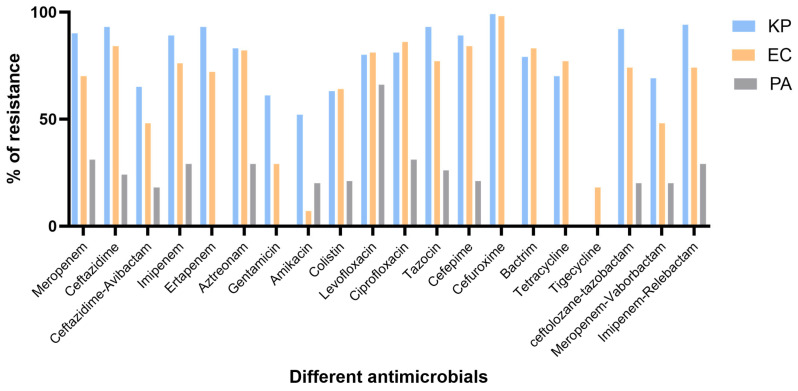
Resistance rates of the different bacterial species against different antimicrobials. AST against tigecycline was performed exclusively for *E. coli* isolates. In contrast, ertapenem, cefuroxime, sulphamethoxazole-trimethoprim (Bactrim), gentamicin, and tetracycline were not recommended for *P. aeruginosa* isolates. KP: *Klebsiella pneumoniae*; PA: *P. aeruginosa*; EC: *Escherichia coli*.

**Figure 2 microorganisms-13-02015-f002:**
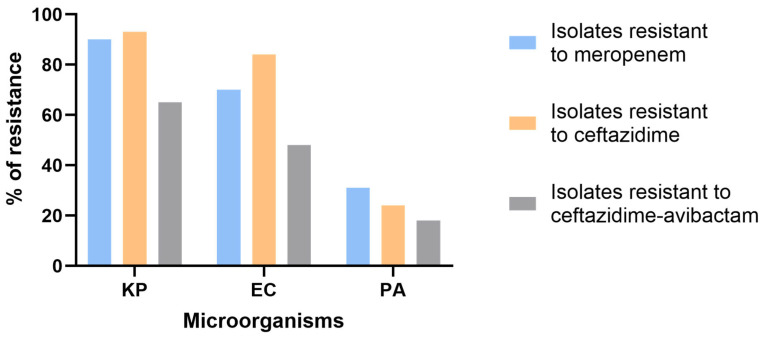
Percentage of isolates resistant to meropenem, ceftazidime, and ceftazidime–avibactam. KP: *Klebsiella pneumoniae*; PA: *P. aeruginosa*; EC: *Escherichia coli*.

**Figure 3 microorganisms-13-02015-f003:**
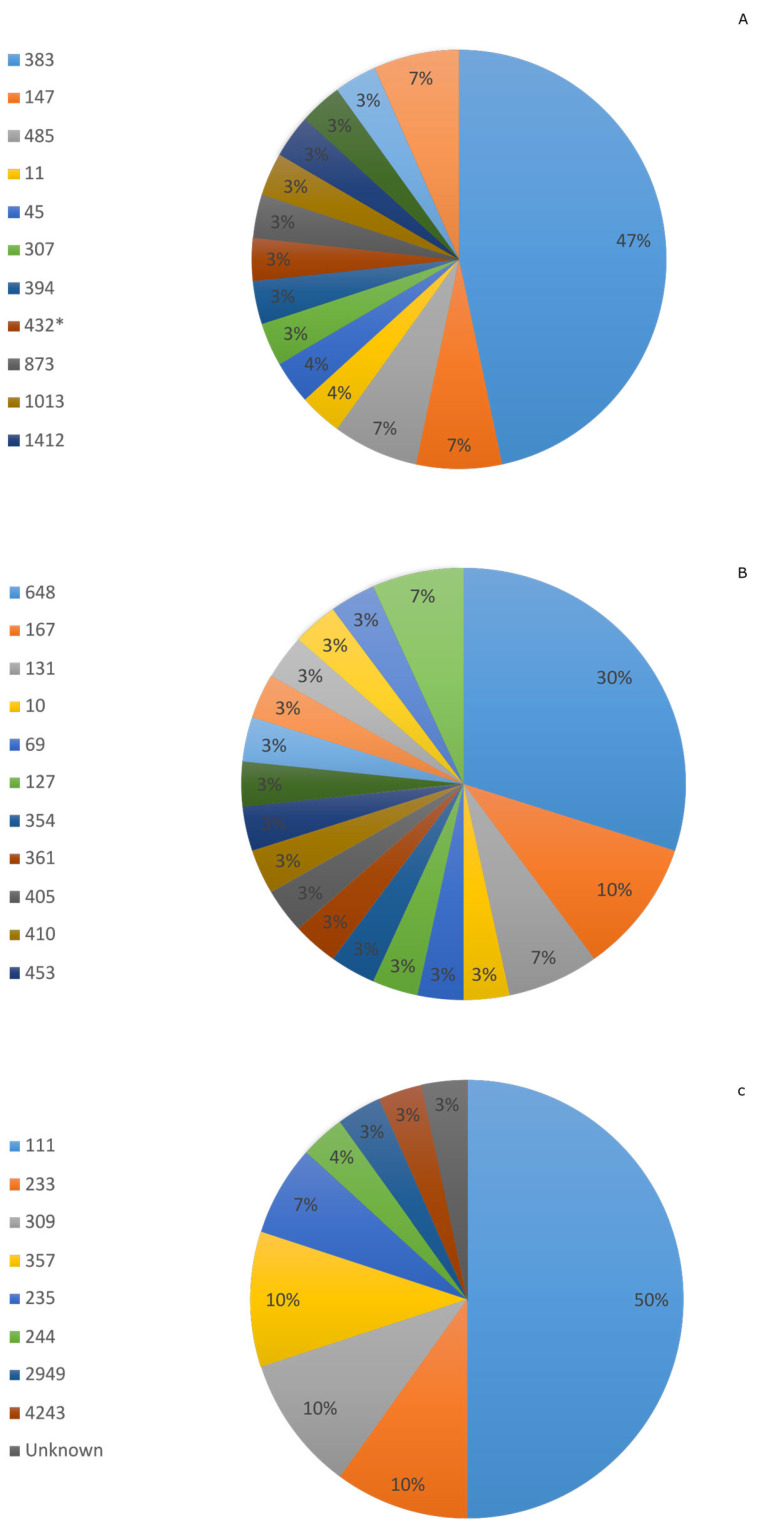
Distribution of sequence types (STs) among clinical isolates of (**A**) *Klebsiella pneumoniae*, (**B**) *Escherichia coli*, and (**C**) *Pseudomonas aeruginosa*; * is provisional or unassigned sequence type.

**Figure 4 microorganisms-13-02015-f004:**
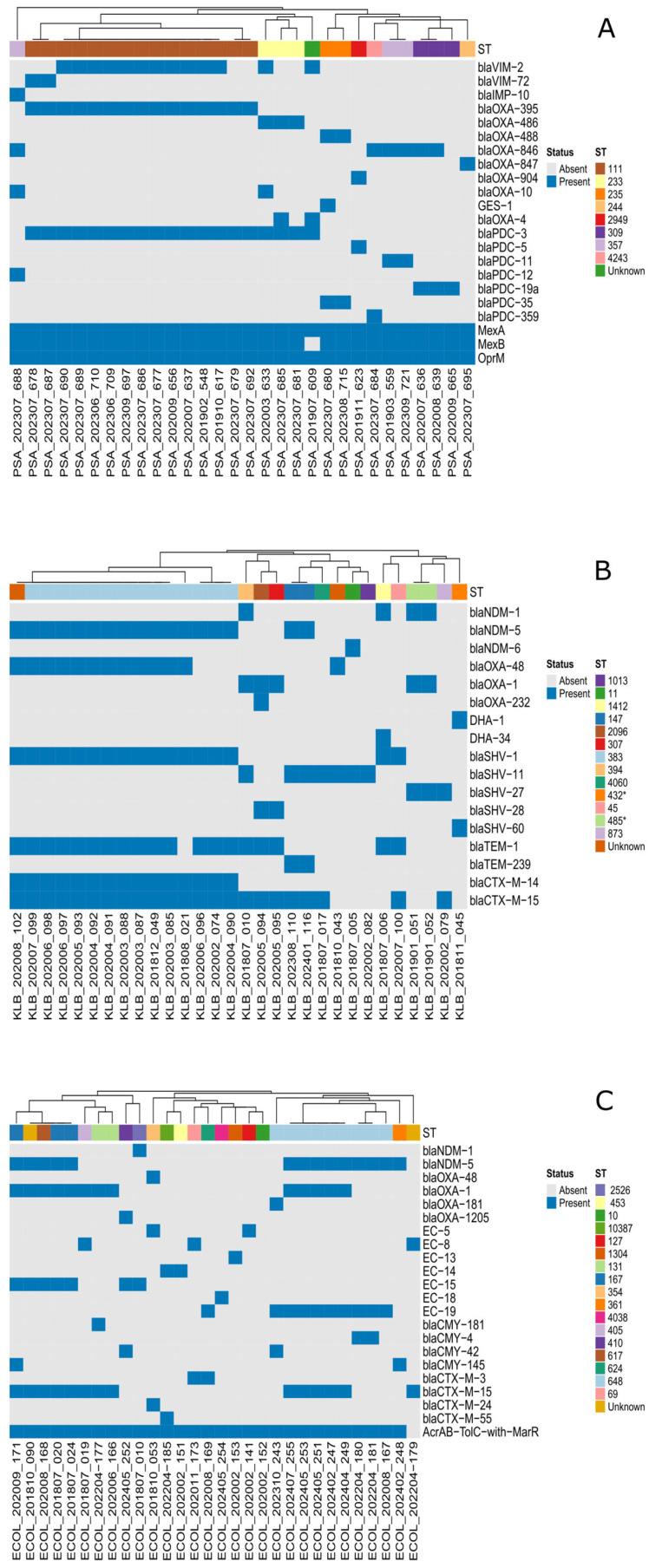
Heatmap representing the distribution of key antimicrobial resistance genes across different isolates of various sequence types. The AMR genes represented contribute to resistance against beta-lactams, cephalosporins, cephamycins, and carbapenems, as well as ceftazidime–avibactam. (**A**) *P. aeruginosa* sequenced isolates, (**B**) *K. pneumoniae* sequenced isolates, (**C**) *Escherichia coli* sequenced isolates.

**Figure 5 microorganisms-13-02015-f005:**
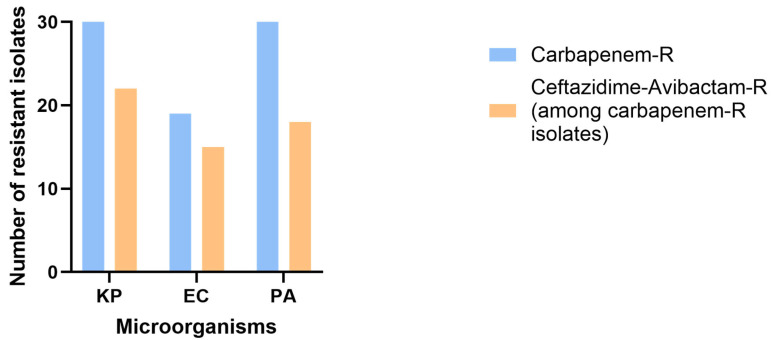
Percentage of sequenced isolates resistant to carbapenems and ceftazidime–avibactam. KP: *Klebsiella pneumoniae*; PA: *P. aeruginosa*; EC: *Escherichia coli*.

**Figure 6 microorganisms-13-02015-f006:**
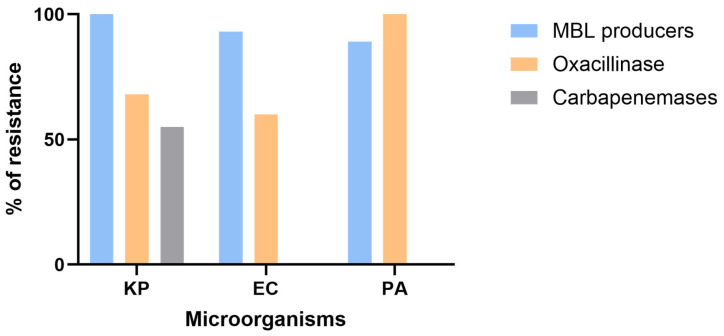
Percentage of CAZ-AVI resistant isolates encoding MBLs, oxacillinases, and OXA-type carbapenemases. KP: *Klebsiella pneumoniae*; PA: *Pseudomonas aeruginosa*; EC: *Escherichia coli;* MBL: metallo-β-lactamase; OXA: oxacillinase.

## Data Availability

The original contributions presented in this study are included in the article/Appendix A. Further inquiries can be directed to the corresponding author.

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
