# Peer review of "Molecular Epidemiology of Carbapenem-Resistant P. aeruginosa and Enterobacterales Isolates from Clinical Infections and Their Susceptibility to Ceftazidime–Avibactam"

_microorganisms, 2025, doi:10.3390/microorganisms13092015_

Round 1
Reviewer 1 Report
Comments and Suggestions for Authors
In this manuscript, the authors present carbepenem - resistance, ceftazidime-avibactam-resistance and presence of resistance genes in a collection of clinical P. aeruginosa, K. pneumoniae, E. coli isolates. The authors describe relatively high rates of ceftazidime-avibactam-resistance in carbapenem-resistant isolates
The manuscript is well written and results are clearly presented. Nevertheless, the number of isolates are relatively low and only three isolates had other than the common resistance mechanisms (MBL). Perhaps, the authors could enrich the manuscript by adding information on carbapenem resistance (or involved genes, NDM or VIM) in Enterobacterales and P. aeruginosa in their settings or in their area.
Figure 5 is misleading. I would advise to use number of isolates (not percentages) or delete it. most of the information is also presented in Figure 2
Author Response
Comment 1: The manuscript is well written and results are clearly presented. Nevertheless, the number of isolates are relatively low and only three isolates had other than the common resistance mechanisms (MBL). Perhaps, the authors could enrich the manuscript by adding information on carbapenem resistance (or involved genes, NDM or VIM) in Enterobacterales and P. aeruginosa in their settings or in their area.
Response 1: The discussion is now adjusted to include a comparison between the trends observed in our study and the regional data. The added paragraph in the discussion reads as follows: “ Our findings align with previous reports from Lebanon and the Middle East, where blaNDM was the most predominant MBL in carbapenem-resitant Enterobacterales and blaVIM in Pseudomonas aeruginosa [22, 23]. For example, Sobh et al. documented the spread of NDM-5 as the most common carbapenemase in Enterobacterales while Al-alaq et al. reported VIM-1 carrying P. aeruginosa isolated from Lebanon. Moreover, similar patterns were reported in Bahrain, Saudi Arabia and Egypt, where NDM-producing K. pneumonia and VIM-producing P. aeruginosa dominate the carbapenem-resistant Gram-negative landscape [24, 25, 26]. These trends are mirrored in our study where CAZ-AVI-resistance is due to MBL expression. This consistent pattern across studies highlights the sustained circulation of these high-risk carbapenemase genes in the whole region.”
Comment 2: Figure 5 is misleading. I would advise to use number of isolates (not percentages) or delete it. most of the information is also presented in Figure 2.
Response 2: Figure 5 label is converted from percentage to numbers. Please note that this figure is different from figure 2 since it only discusses the isolates that were sequenced (90) and not the total isolates (568).
Reviewer 2 Report
Comments and Suggestions for Authors
- I appreciate the opportunity to review this manuscript, as infections caused by Pseudomonas aeruginosa and Enterobacteriaceae are of great importance worldwide. I would like to make some comments and suggestions that could improve the quality of the manuscript.
- Title: The title is confusing, as it seems like they're going to analyze the molecular epidemiology of infections, but in reality, they're going to analyze Enterobacteriaceae and Pseudomonas strains through molecular epidemiology. I suggest changing the title, as it's somewhat confusing.
- Introduction: On line 58, the Enterobacteriaceae family should be in italics.
- Line 63: From what I see, they only describe the origin of the NDM1 beta-lactamase, but not the origin of the other beta-lactamases. I suggest including the meaning of each acronym.
- In general, although the introduction presents the state of the art of antimicrobial resistance, I believe that the problem of antimicrobial resistance must be clearly identified and linked to the COVID-19 pandemic. According to the literature, Pseudomonas and Enterobacteriaceae caused an additional problem in COVID-19 patients. There are several studies that discuss outbreaks caused by these microorganisms in critically ill COVID-19 patients.
- I consider the topic of antimicrobial resistance to be relevant, and consequently the introduction must be more extensive and provide additional richness that gives value to this section, since it is limited to less than one page, of course, with the inclusion of new bibliography.
- Materials and Methods: Just as a question, don't you have a classification of the source of isolation for each of the test microorganisms? I understand they're part of a collection, but if you had information about their origin, that would be ideal. It's not mandatory.
- Line 92: From what I see, you're describing the standardized method for the disk diffusion test and using the CLSI interpretation guidelines. Was the selection of antibiotics based on a consensus on resistance classification? This is to classify microorganisms on the antimicrobial resistance spectrum, that is, multidrug-resistant, extreme drug-resistant, and pan-drug-resistant. It would be appropriate to classify the isolates, although it's not mandatory; it's merely a suggestion.
- I understand that two methods were used to determine antimicrobial resistance profiles. But what was the criterion for using two different methods? Was any correlation analysis performed between tests? Or was there some exercise that would allow for parallel interpretation of both methods? If not, I think one of the methods would be redundant, since both are recognized by international standards, but as far as I understand, the cutoff points are different. Please clarify this question.
- Line 105: According to this section, only 30 representative isolates from each genus and species were selected. What was the criterion for selecting this number? Was it based on antimicrobial resistance findings? Did these strains show different resistance profiles that warranted whole-genome sequencing? If so, a paragraph clarifying this question should be included, since it appears that the selection of isolates for whole-genome sequencing was arbitrary.
- Line 141: From what I see there, the classification criteria for MDR, XDR, and PDR are described. So, they used a classification consensus. That should be described in the materials and methods section. What was the classification criterion or consensus?
- Figure 1: It is noteworthy that the resistance percentage axis has a maximum of 150. I suggest correcting the axis, as well as improving the quality, and I suggest removing the title from the graph. By definition, any figure must be described at the bottom of it, as well as the title.
- Figure 1: The acronyms KP, EC, PA, must be described in the figure.
- I recommend that, given this very valuable information, these resistance profiles could be presented in another way, possibly as heat maps. This is a recommendation, not a mandatory one.
- Figure 2: It is important for authors to keep in mind that graphs must be of publication quality and must be interpreted individually. I recommend describing the acronyms for the microorganisms in the figure. They also indicate the number of isolates resistant to each antibiotic, but they are reporting frequencies as a percentage. Authors must define whether this represents the number of isolates or the percentage of isolates.
- Line 165: The authors do not indicate why they chose 30 isolates of each bacterial species. I suggest a solid justification; I assume it's due to the resistance profiles; they chose the most relevant resistance profiles for whole-genome analysis. Please clarify.
- Figure 3: The figure title is very ambiguous and lacks a description that allows for a comprehensive understanding. I suggest rewording the figure title. I also encourage authors to improve the quality of the figures, as they are not clearly visible.
- Figure 4: I consider the information presented to be very valuable, but it lacks a comprehensive description. I understand that they want to simultaneously show the presence or absence of resistance genes along with the type sequences, but I think they should be presented in a larger format and even spread out on a single page.
- Figure 5 and 6. The authors recommend that the axes be indicated with a scale up to 100%; it is not usual for the graphs to be indicated with a maximum percentage of 150. On the other hand, the acronyms of each microorganism must have their meaning indicated on one of the axes. They indicate the word organisms, when in reality they are microorganisms.
- Line 213: The names of microorganisms are not italicized.
- Line 224: Although the authors found variants in some resistance genes, it would be interesting to identify whether there was a correlation between the detected variant and resistance to the test antibiotics to see if there is a direct relationship between the phenotype and the detected genotype.
- Line 257: I consider the word "arsenal" to be somewhat premeditated and informal to indicate that a microorganism has several resistance mechanisms. I understand that if the authors used genome sequencing technologies, the more appropriate term would be "resistome," to refer to the set of resistance determinants that each microorganism exhibits.
- Line 284: The authors discuss the presence of the identified type sequences in the collection of microorganisms tested. It would be interesting to know what criteria the authors considered in classifying each of the type sequences. That is, which constitutive genes were analyzed to make the corresponding assignment, since this information is not described in the Materials and Methods section.
- Lines 289 to 293: The authors attempt to discuss the presence of plasmids and the ability of E. coli to carry out horizontal genetic transfer. However, to support this hypothesis, it is important that the plasmid findings be accompanied by a search for important elements that are crucially involved in horizontal genetic transfer, such as the presence of "tra" operons. These operons are directly related to the minimal machinery necessary for the host bacterium to carry out horizontal genetic transfer mediated through plasmids.
- Conclusion: By definition, a conclusion must be consistent with the findings presented in the manuscript. The conclusion I read highlights the problem of high levels of resistance to some of the antibiotics tested, but there are no conclusions related to the specific findings described in the materials and methods, such as the presence of type sequences or the identified genetic resistance elements. You should reformulate your conclusion and align it with the laboratory findings.
Author Response
Comment 1: Title: The title is confusing, as it seems like they're going to analyze the molecular epidemiology of infections, but in reality, they're going to analyze Enterobacteriaceae and Pseudomonas strains through molecular epidemiology. I suggest changing the title, as it's somewhat confusing:
Response 1: The title is changed to “Molecular Epidemiology of Carbapenem-Resistant aeruginosa and Enterobacterales Isolates from Clinical Infections and Their Susceptibility to Ceftazidime-Avibactam”, this way the focus is shifted from analyzing infections to analyzing strains through molecular epidemiology.
Comment 2: Introduction: On line 58, the Enterobacteriaceae family should be in italics:
Response 2: All Enterobacteriaceae and Enterobacterales are formatted accordingly.
Comment 3: Line 63: From what I see, they only describe the origin of the NDM1 beta-lactamase, but not the origin of the other beta-lactamases. I suggest including the meaning of each acronym:
Response 3: The paragraph is formatted to read as follows: Unfortunately, this increased utilization of carbapenems has contributed to the development of resistance to these antibiotics in Pseudomonas aeruginosa (CRPA) and Enterobacterales (CRE) through various mechanisms such as OXA-48-like family (class D Oxacillinases) with carbapenem-hydrolizing activity, Klebsiella pneumoniae carbapenemases (KPCs), New Delhi Metallo-β-lactamase (NDM), Imipenemase (IMP), and Verona integron-encoded metallo-β-lactamase (VIM) [8,9]. Additionally, resistance can occur through the loss of porins and up-regulation of efflux pumps [10,11].
Comment 4: In general, although the introduction presents the state of the art of antimicrobial resistance, I believe that the problem of antimicrobial resistance must be clearly identified and linked to the COVID-19 pandemic. According to the literature, Pseudomonas and Enterobacteriaceae caused an additional problem in COVID-19 patients. There are several studies that discuss outbreaks caused by these microorganisms in critically ill COVID-19 patients:
Response 4: While we acknowledge that, the COVID-19 pandemic has had a significant impact on antimicrobial resistance patterns globally—particularly in hospital settings—our study was not designed to assess this association. The isolate collection began in 2017 and spans both pre- and post-pandemic periods; however, we do not have metadata on patients’ COVID-19 status, and therefore could not evaluate the specific influence of the pandemic on the isolates studied. For this reason, we chose not to directly address COVID-19 in our analysis. Moreover, we added in the materials and methods section a clause clearly defining the collection date of the isolates “A total of 394 aeruginosa, 90 E. coli and 84 K. pneumoniae isolates, collected between 2017 and 2024."
Comment 5: I consider the topic of antimicrobial resistance to be relevant, and consequently the introduction must be more extensive and provide additional richness that gives value to this section, since it is limited to less than one page, of course, with the inclusion of new bibliography.
Response 5: The introduction is now altered and enriched.
Comment 6: Materials and Methods: Just as a question, don't you have a classification of the source of isolation for each of the test microorganisms? I understand they're part of a collection, but if you had information about their origin, that would be ideal. It's not mandatory:
Response 6: The isolates used in this study are part of a long-standing institutional collection. Unfortunately, detailed metadata regarding the exact source of isolation (e.g., clinical specimen type) is not available for all isolates. However, for those with known origins, the sources primarily include blood, urine, trachea, sputum, DTA, wound, swabs and bone samples. For this reason, we did not include source classification in the manuscript.
Comment 7: Line 92: From what I see, you're describing the standardized method for the disk diffusion test and using the CLSI interpretation guidelines. Was the selection of antibiotics based on a consensus on resistance classification? This is to classify microorganisms on the antimicrobial resistance spectrum, that is, multidrug-resistant, extreme drug-resistant, and pan-drug-resistant. It would be appropriate to classify the isolates, although it's not mandatory; it's merely a suggestion.
Response 7: The selection of antibiotics was guided by the international consensus definitions proposed by Magiorakos et al. (2012) to allow for classification of isolates as multidrug-resistant (MDR), extensively drug-resistant (XDR), or pandrug-resistant (PDR). This classification was based on susceptibility profiles interpreted using CLSI breakpoints and has now been clearly described in the Materials and Methods section.
Comment 8: I understand that two methods were used to determine antimicrobial resistance profiles. But what was the criterion for using two different methods? Was any correlation analysis performed between tests? Or was there some exercise that would allow for parallel interpretation of both methods? If not, I think one of the methods would be redundant, since both are recognized by international standards, but as far as I understand, the cutoff points are different. Please clarify this question: In our study, we employed both disk diffusion and broth microdilution for antimicrobial susceptibility testing. The rationale for using both was based primarily on resource availability and method-specific suitability:
Comment 9: Disk diffusion was used as the routine method for a broad panel of antibiotics, following CLSI guidelines, due to its feasibility and cost-effectiveness in our setting. However, broth microdilution was applied for selected antibiotics such as colistin, for which disk diffusion is not reliable or not recommended due to poor diffusion characteristics.
Response 9: Due to the difference in testing purpose and antibiotic selection, no direct correlation or categorical agreement analysis was performed between the two methods in this study, as they were used selectively rather than in parallel for all agents. However, each method was interpreted according to CLSI breakpoints appropriate for that method, and results were used complementarily.
Clarification was added to the materials and methods section as follows: “The choice of method was based on resource availability and the suitability of the method for specific antibiotics. Disk diffusion was used as the standard approach for the majority of antibiotics due to its cost-effectiveness and accessibility in our routine laboratory setting. Broth microdilution method were employed for antibiotics, such as colistin, where disk diffusion is not reliable or not recommended due to poor diffusion characteristics. No formal comparison or correlation analysis was performed between the two methods, as they were not applied in parallel for all agents but rather in a complementary manner.”
Comment 10: Line 105: According to this section, only 30 representative isolates from each genus and species were selected. What was the criterion for selecting this number? Was it based on antimicrobial resistance findings? Did these strains show different resistance profiles that warranted whole-genome sequencing? If so, a paragraph clarifying this question should be included, since it appears that the selection of isolates for whole-genome sequencing was arbitrary:
Response 10: Thank you for pointing this out. The selection of 30 representative isolates per species for whole-genome sequencing was based on the need to capture a diverse range of antimicrobial resistance profiles and clinical backgrounds. Isolates were chosen to reflect variability in phenotypic resistance patterns (including MDR and XDR profiles), and unique or atypical AST results that warranted further genomic investigation. While the number 30 was also influenced by practical considerations such as cost and sequencing capacity, care was taken to ensure that the subset was both epidemiologically and phenotypically representative of the broader collection. We have revised the manuscript to clarify this selection rationale and it reads as follows: “The selection was based on phenotypic diversity, including a range of antimicrobial re-sistance profiles (from susceptible to MDR, XDR), and unique resistance patterns that merited further genomic investigation. This selection aimed to reflect the broader collection while also considering feasibility and cost limitations.”
Comment 11: Line 141: From what I see there, the classification criteria for MDR, XDR, and PDR are described. So, they used a classification consensus. That should be described in the materials and methods section. What was the classification criterion or consensus?
Response 11: We have now explicitly stated in the Materials and Methods section that classification was performed according to the international consensus definitions proposed by Magiorakos et al. (2012), using susceptibility profiles interpreted based on CLSI breakpoints, the paragraph reads as follows: “Multidrug-resistant (MDR), extensively drug-resistant (XDR), and pandrug-resistant (PDR) classifications were defined according to the international consensus criteria proposed by Magiorakos et al. (2012), using susceptibility profiles interpreted based on CLSI breakpoints. MDR was defined as non-susceptibility to at least one agent in three or more antimicrobial categories; XDR as non-susceptibility to at least one agent in all but two or fewer categories; and PDR as non-susceptibility to all agents in all categories. Antimicrobial categories were selected based on the list provided in the original publication and adapted to the antibiotics tested in our study, ensuring coverage of the major classes outlined in the consensus definitions. “
We also ensured that MDR and XDR classifications were revised for consistency across all relevant categories, including the correction for previously missing fosfomycin data in E. coli.
Comment 12: Figure 1: It is noteworthy that the resistance percentage axis has a maximum of 150. I suggest correcting the axis, as well as improving the quality, and I suggest removing the title from the graph. By definition, any figure must be described at the bottom of it, as well as the title:
Response 12: All axis percentages have been set to 100.
Comment 13: Figure 1: The acronyms KP, EC, PA, must be described in the figure:
Response 13: All acronyms are described among all figures,
Comment 14: I recommend that, given this very valuable information, these resistance profiles could be presented in another way, possibly as heat maps. This is a recommendation, not a mandatory one:
Response 14: We agree that heat maps provide a clear and intuitive visualization of antimicrobial susceptibility patterns across isolates. Due to space limitations and the large number of isolates included in our study, we have focused on summary data and key analyses in the main manuscript.
Comment 15: Figure 2: It is important for authors to keep in mind that graphs must be of publication quality and must be interpreted individually. I recommend describing the acronyms for the microorganisms in the figure. They also indicate the number of isolates resistant to each antibiotic, but they are reporting frequencies as a percentage. Authors must define whether this represents the number of isolates or the percentage of isolates:
Response 15: We have revised the figure to ensure publication-quality formatting and have added full descriptions of the microorganism acronyms in the figure legend for clarity. Additionally, we have clarified in both the figure and legend that the data presented represent the percentage of resistant isolates, not absolute numbers. This has been clearly defined to avoid any ambiguity in interpretation.
Comment 16: Line 165: The authors do not indicate why they chose 30 isolates of each bacterial species. I suggest a solid justification; I assume it's due to the resistance profiles; they chose the most relevant resistance profiles for whole-genome analysis. Please clarify:
Response 16: As discussed earlier, we selected 30 representative isolates from each bacterial species was based on the need to capture a diverse range of antimicrobial resistance profiles and clinical backgrounds. This approach was intended to ensure a comprehensive representation of the genetic diversity and resistance mechanisms within each species for whole-genome sequencing analysis.
Comment 17: Figure 3: The figure title is very ambiguous and lacks a description that allows for a comprehensive understanding. I suggest rewording the figure title. I also encourage authors to improve the quality of the figures, as they are not clearly visible:
Response 17: We have revised the title of Figure 3 to clearly indicate that it presents the distribution of sequence types (STs) among pneumoniae, P. aeruginosa, and E. coli clinical isolates. In addition, the figure has been improved for better clarity and resolution in line with publication standards.
Comment 18: Figure 4: I consider the information presented to be very valuable, but it lacks a comprehensive description. I understand that they want to simultaneously show the presence or absence of resistance genes along with the type sequences, but I think they should be presented in a larger format and even spread out on a single page:
Response 18: We thank the reviewer for recognizing the value of the data presented in Figure 4. In response to the comment, we have revised the figure legend to provide a more comprehensive and clearer description of the data and its interpretation. The legend reads as follows: “Heatmap representing the distribution of key antimicrobial resistance genes across different isolates of various sequence types. The AMR genes represented contribute to resistance against: beta-lactams, cephalosporins, cephamycins, carbapenems as well as ceftazidime-avibactam.” Additionally, we submitted a higher-resolution version of all the figures as recommended, to enhance readability and clarity.
Comment 19: Figure 5 and 6. The authors recommend that the axes be indicated with a scale up to 100%; it is not usual for the graphs to be indicated with a maximum percentage of 150. On the other hand, the acronyms of each microorganism must have their meaning indicated on one of the axes. They indicate the word organisms, when in reality they are microorganisms:
Response 19: In Figures 5 and 6, we have now adjusted the y-axis scale to display a maximum of 100%, which aligns with standard conventions for percentage-based data. Additionally, we have clarified the meaning of the acronyms used for each bacterial species in the figure legend. The terminology in the figure and manuscript has also been updated to refer to “microorganisms” instead of “organisms” to ensure scientific accuracy.
Comment 20: Line 213: The names of microorganisms are not italicized:
Response 20: The names of the microorganisms are italicized throughout the manuscript.
Comment 21: Line 224: Although the authors found variants in some resistance genes, it would be interesting to identify whether there was a correlation between the detected variant and resistance to the test antibiotics to see if there is a direct relationship between the phenotype and the detected genotype:
Response 21: As discussed in the subsection 3.4, the isolates from which these variants were found had no MBL genes yet found to be resistant to CAZ-AVI. We changed the paragraph to clearly highlight the latter: “ We found single nucleotide variants (SNVs) in ampD (D183Y, G148A, R11L) and ampR (D135N) for CAZ-AVI resistant PSA_202307_695. As for PSA_202309_721 isolate, SNVs were detected in mexR (V126E) and in ampD (A96T, E68D); in ampR, SNVs (M288R, S179T, E114A) and substitutions (RG282RE), all resulting in missense mutations contributing to phenotypic resistance to CAZ-AVI. For the coli isolate, we identified various amino acid alterations: in ampC, SNVs (D367A, D304G, L257R, A157T), and substitutions (RP312HP, NE260ND, LK254MN, VQ250VR, PPA208PPP); in marR, SNVs (G103S, Y137H), all resulting in missense mutations; in ompC, substitutions (SLA295SVA, GAI216AAV, GNPSG175GSVSG) resulting in missense mutations, and highly disruptive INDELs (RG308TIAGRN, FTSGVT182MT) leading to frameshift mutations (Table S1). This finding suggests a possible link between the observed genotype and CAZ-AVI resistance.”
Moreover, in the discussion, we highlighted by referring to other studies on the role played by these specific mutation in terms of resistance to CAZ-AVI: “As no MBL genes were detected in the remaining CAZ-AVI-resistant P. aeruginosa and E. coli isolates, a detailed genomic analysis was conducted, revealing that CAZ-AVI resistance is multifactorial and attributed to other mechanisms. In P. aeruginosa isolates, resistance is likely driven by mutations in ampD and ampR, leading to AmpC hyperproduction [21], and in mexR, causing overexpression of the MexAB-OprM efflux pump, which slightly raises the MIC of CAZ-AVI [22,23,24]. For the E. coli isolate, resistance is primarily due to ompC frameshift INDELs severely reducing outer membrane permeability, a critical factor given CAZ-AVI's reliance on passive diffusion for cellular entry [25]. Additionally, marR mutations enhance AcrAB-TolC efflux, and numerous ampC substitutions potentially alter beta-lactamase activity. These combined chromosomal alterations provide a robust mechanism for evading CAZ-AVI activity [26].”
Comment 22: Line 257: I consider the word "arsenal" to be somewhat premeditated and informal to indicate that a microorganism has several resistance mechanisms. I understand that if the authors used genome sequencing technologies, the more appropriate term would be "resistome," to refer to the set of resistance determinants that each microorganism exhibits.
Response 22: The word arsenal was switched to resistome for a better scientific clarity. The sentence reads as follows: “Moreover, using WGS, we uncovered the molecular mechanisms of resistance of these isolates by revealing their resistomes.”
Comment 23: Line 284: The authors discuss the presence of the identified type sequences in the collection of microorganisms tested. It would be interesting to know what criteria the authors considered in classifying each of the type sequences. That is, which constitutive genes were analyzed to make the corresponding assignment, since this information is not described in the Materials and Methods section:
Response 23: We have now added this information to the Materials and Methods section (2.7) of the manuscript, the section reads as follows: “Sequence types were assigned using multilocus sequence typing (MLST) schemes specific to each species via the MLST tool from CGE. For pneumoniae, the scheme included the housekeeping genes gapA, infB, mdh, pgi, phoE, rpoB, and tonB. For E. coli, the genes analyzed were adk, fumC, gyrB, icd, mdh, purA, and recA as per the Achtman MLST scheme. For P. aeruginosa, we used the standard scheme targeting acsA, aroE, guaA, mutL, nuoD, ppsA, and trpE. Assignments were cross-validated using the PubMLST database. (https://cge.food.dtu.dk/services/MLST/).”
Comment 24: Lines 289 to 293: The authors attempt to discuss the presence of plasmids and the ability of E. coli to carry out horizontal genetic transfer. However, to support this hypothesis, it is important that the plasmid findings be accompanied by a search for important elements that are crucially involved in horizontal genetic transfer, such as the presence of "tra" operons. These operons are directly related to the minimal machinery necessary for the host bacterium to carry out horizontal genetic transfer mediated through plasmids.
Response 24: In this study, our objective was not to explore the molecular mechanisms underlying horizontal gene transfer, such as the presence of tra operons or other conjugation-associated machinery. Rather, the focus of our analysis was to describe the distribution of plasmid replicons and antimicrobial resistance genes across the coli isolates using whole-genome sequencing, with the aim of characterizing the epidemiological features and resistance determinants present in our collection. While we acknowledge that screening for tra operons could provide deeper insights into the mobility of plasmids, such an investigation falls outside the scope of our current study. To further clarify the latter, the modified discussion part now reads as follows: “Furthermore, the presence of multiple plasmids, such as IncHI1B(pNDM-MAR) and IncFIB(pNDM-Mar) in K. pneumoniae, widely associated with blaNDM and blaOXA-48-like carbapenemase genes, suggesting their role in the dissemination of carbapenem resistance. Moreover, IncFIA and IncFIB (AP001918) present in E. coli were frequently found in isolates carrying blaNDM, suggesting a potential for horizontal gene transfer leading to the spread of carbapenem-resistance genes [32, 33].
Comment 25: Conclusion: By definition, a conclusion must be consistent with the findings presented in the manuscript. The conclusion I read highlights the problem of high levels of resistance to some of the antibiotics tested, but there are no conclusions related to the specific findings described in the materials and methods, such as the presence of type sequences or the identified genetic resistance elements. You should reformulate your conclusion and align it with the laboratory findings.
Response 25: The conclusion is now altered to highlight the results presented in the manuscript, it now reads as follows: “In conclusion, our findings reveal alarmingly high levels of resistance to CAZ-AVI among carbapenem-resistant pneumoniae, E. coli and P. aeruginosa isolates. Whole-genome sequencing revealed a high diversity of sequence types and a wide variety of acquired resistance genes contributing to both carbapenem and ceftazidime-avibactam resistance. These results emphasize the critical importance of molecular diagnostics, antimicrobial stewardship, and collaborative surveillance to guide effective therapeutic strategies and mitigate the spread of CAZ-AVI resistance in high-burden settings. In the short term, combining other antimicrobial agents such as aztreonam with CAZ-AVI may offer a viable option against resistant isolates until novel antimicrobials are developed.”
Round 2
Reviewer 2 Report
Comments and Suggestions for Authors
The comments and suggestions regarding your manuscript have been satisfactorily addressed. I consider that it can be approved for publication in its current form.